# The Phytochemical Analysis of *Vinca* L. Species Leaf Extracts Is Correlated with the Antioxidant, Antibacterial, and Antitumor Effects

**DOI:** 10.3390/molecules26103040

**Published:** 2021-05-19

**Authors:** Alexandra Ciorîță, Cezara Zăgrean-Tuza, Augustin C. Moț, Rahela Carpa, Marcel Pârvu

**Affiliations:** 1Faculty of Biology and Geology, Babeș-Bolyai University, 44 Republicii St., 400015 Cluj-Napoca, Romania; rahela.carpa@ubbcluj.ro (R.C.); marcel.parvu@ubbcluj.ro (M.P.); 2National Institute for Research and Development of Isotopic and Molecular Technologies, 67-103 Donath St., 400293 Cluj-Napoca, Romania; 3Faculty of Chemistry and Chemical Engineering, Babeș-Bolyai University, 11 Arany János St., 400028 Cluj-Napoca, Romania; cezarazt7@gmail.com (C.Z.-T.); augustin.mot@ubbcluj.ro (A.C.M.)

**Keywords:** alkaloids, vincamine, antioxidant, antibacterial, phenolic compounds

## Abstract

The phytochemical analysis of *Vinca minor*, *V. herbacea*, *V. major*, and *V. major* var. *variegata* leaf extracts showed species-dependent antioxidant, antibacterial, and cytotoxic effects correlated with the identified phytoconstituents. Vincamine was present in *V. minor*, *V. major*, and *V. major* var. *variegata*, while *V. minor* had the richest alkaloid content, followed by *V. herbacea*. *V. major* var. *variegata* was richest in flavonoids and the highest total phenolic content was found in *V. herbacea* which also had elevated levels of rutin. Consequently, *V. herbacea* had the highest antioxidant activity followed by *V. major* var. *variegata*. Whereas, the lowest one was of *V. major*. The *V. minor* extract showed the most efficient inhibitory effect against both *Staphylococcus*
*aureus* and *E. coli*. On the other hand, *V. herbacea* had a good anti-bacterial potential only against *S. aureus*, which was most affected at morphological levels, as indicated by scanning electron microscopy. The *Vinca* extracts acted in a dose-depended manner against HaCaT keratinocytes and A375 melanoma cells and moreover, with effects on the ultrastructure, nitric oxide concentration, and lactate dehydrogenase release. Therefore, the *Vinca* species could be exploited further for the development of alternative treatments in bacterial infections or as anticancer adjuvants.

## 1. Introduction

The progress of modern medicine in the last century is the result of multiple, relentless battles fought against environmental and inborn threats against humans. The diseases associated with these harmful factors had driven the efforts of researchers and physicians toward the modern medicine practices, that are frequently updated. Antibiotic-resistant bacteria and invasive malignant tumors are two of the most challenging, contemporary problems that scientists are constantly attempting to overcome [1,2]. Since the gene for antibiotic resistance in bacteria was identified to be present also in an ancient strain [3], far before antibiotics were even discovered, resources were increasingly directed towards alternative means of treatment [4,5]. At the same time, chemotherapeutic drugs remain the main treatment administered in cancer patients, even if the side effects are well-known and thoroughly documented [6]. Given the above-mentioned difficulties that are encountered along with the evolution of therapeutics, scientists concentrated their attention on phytotherapy [7].

Plants are the foundation of modern medicine, as they synthesize natural products with effective antibacterial [8], antifungal [9], antiparasitic [10], and antitumor [11] activities. Among these medicinal plants are also the *Vinca* species. The United States Department of Agriculture [12] describes three species and one variety for the *Vinca* genus: *V. major* L., *V. minor* L., *V. herbacea* Walds. and Kit., and *V. major* L. var. *variegata* ‘Louden’. The European Environment Agency [13] includes several *V. major* synonyms (*V. erecta* Regel and Schmalh, *V. pubescens* d’Urv) in their database and treats *V. difformis* Pourret as an independent species. In his work, Koyuncu (2012) identified and described a new species (*Vinca soneri* Koyuncu) [14], which is currently included only in the Plants of the World database [15].

These plants had been intensely studied for their therapeutic properties [16,17], due to the rich alkaloid content [18,19,20,21]. Moreover, a relation between the morphological and ultrastructural aspects of the leaf in *Vinca* species, and the chemical composition is yet to be established [22]. *Vinca* alkaloids belong to the terpene indole type with more than 50 described only for *V. minor* [23,24]. Distinct types of *Vinca* alkaloids have similar structures, but their mode of action and toxicological profiles vary extensively [25].

The first isolation of a *Vinca* alkaloid was made by Lucas in 1859, and since then the research continued along the centuries [26]. In the early 1900′s, *Vinca* species were still insufficiently studied, and three new alkaloids were described for *V. minor* and three for *V. herbacea* [27,28,29]. Later, 43 alkaloids were isolated from *V. major*, 40 from *V. minor* and three from *V. herbacea*. After 2010 and up until now, new alkaloids are discovered and isolated almost every year only for *V. minor*, nine new alkaloids were described and isolated from *V. major* and two from *V. herbacea* [19,20,27,30,31].

One of the most important alkaloids of the *Vinca* species is vincamine [32] recognized as one of the few alkaloids that has beneficial effects on living cells [33]. Unlike other alkaloids found in the *Vinca* species analyzed herein, vincamine is commercially available and included in dietary supplements, but the currently used active compound (vinpocetine) is a semi-synthetic one, derived from vincamine [34,35,36]. Vincamine is most abundant in *V. minor* and acts as a cerebral metabolic enhancer by extending the blood flow and regional glucose uptake, is a neuroprotective against ischemia and hypoxia, and has antioxidant and antiapoptotic properties [37,38]. For example, Wang et al. (2020) designed and synthesized several drugs from vincamine, which inhibited apoptosis in pancreatic β-cells in a dose-dependent manner [39].

Along alkaloids, other important natural compounds found in *Vinca* species are phenolic acids, carotenoids, caffeic acid, iridoids, flavonoids, amino acids, and other phenolic compounds [16,40]. One phenol that seems to stand out in high amounts throughout the *Vinca* family is chlorogenic acid, which is considered as a marker for leaf epidermis metabolites [41]. Other phenols have also been identified in various *Vinca* species, in far lesser concentrations compared to chlorogenic acid, such as *p*-coumaric acid, caffeic acid, ferulic acid, rutin, and kaempferol [42,43].

The aim of this study was to compare the leaf extracts of *Vinca minor*, *V. major*, *V. herbacea*, and *V. major* var. *variegata* in terms of chemical composition, antibacterial activity against *E. coli* and *S. aureus*, and cytotoxic effects against normal keratinocytes (HaCaT) and skin melanoma (A375) human cell lines. Therefore, a correlation between the observed effects in the assayed cells and the phytoconstituents was established, herein, such as: Alkaloids, flavonoids, hydroxybenzoic, or hydroxycinnamic acids. For the first time, these species were investigated together and comparatively, showing that the least familiar *Vinca* species have a great antioxidant, antitumoral, and antibacterial potential and could be exploited further for their therapeutic properties. As a novel approach, the total alkaloid content was determined in the selected species through an adapted Dragendorff assay successfully optimized in this study. Additionally, the morphological effects of the extracts against Gram-negative and Gram-positive bacteria, along with the ultrastructural modifications induced in human cell lines were described using scanning and transmission electron microscopy techniques.

New-generation drugs combine strong antibiotics or cytostatics with natural products found in plants, thus, this study could be the foundation for new research opportunities by exploiting other *Vinca* species with good potential in further, and more advanced investigations.

## 2. Results

### 2.1. Phytochemical Analyses

#### HPLC-DAD Method

HPLC analysis of all four extracts were performed with corresponding chromatograms (Figure 1a). Twenty common standards were chosen, comprising a relatively wide array of plant natural compounds, with representatives in flavonoids, hydroxybenzoic acids, hydroxycinnamic acids, as well as alkaloid groups. Integration of peaks larger than 10 mAbs yielded the results shown in Table 1. All extracts exhibited high levels of chlorogenic acid, whereas *V. herbacea* had a remarkable high content of rutin. Other flavonoids such as quercetin and quercitrin were found, although in lower concentrations. Vincamine was found only in *V. minor*, *V. major*, and *V. major* var. *variegata*.

Each class of compounds had specific UV-Vis absorption features (Figure 1b). Flavonoids displayed an absorption maximum between 340 and 380 nm, hydroxybenzoic acids had peaks with absorption maximum between 230 and 320 nm. Whereas hydroxycinnamic acids had a signature spectrum with a peak between 310 and 350 nm, with a shoulder around 300 nm. Alkaloids had also clearly distinct spectral features with an absorption maximum between 240 and 310 nm.

However, many of the obtained peaks could not be identified. In order to sort this information out, UV-Vis spectra corresponding to all peaks that contributed to more than 1% to the final area were selected for Principal Component Analysis (PCA) after normalization. UV-Vis spectra of all twenty standards were also included in the analysis as a mean of control.

All spectral features (position of absorption maximum, additional absorption wavelengths, spectral shoulders) constituted the bedrock for cluster formation: Each of the four score-plots (Figure 2a–d) exhibited four identical groups. Two of these groups were distinct, namely those attributed to hydroxycinnamic acids and flavonoids. The groups of alkaloids and hydroxybenzoic acids were slightly superimposed. Retention time, spectral features, phytochemical group, and chromatographic area of each identified spectra/peak can be found in Appendix A. Representative spectra of PCA grouped compounds are also shown in Appendix A. Noteworthy, most of the analyzed peaks of unidentified compounds belonged to alkaloid phytochemical group.

For all grouped spectra, the total standard equivalent content was calculated for each class of phytoconstituents identified by PCA analysis (alkaloids, flavonoids, hydroxybenzoic, and hydroxycinnamic acids), as presented in Figure 2e. Alkaloid content as determined by HPLC was highest for *V. herbacea*; *V. major* var. *variegata* and *V. herbacea* were the richest in flavonoids. Data obtained from this set of analyses were further used as a means of comparison for the results obtained by colorimetric determination of total alkaloid content (TAC) and total flavonoid content (TFC).

### 2.2. Colorimetric Determination of Total Phenolic and Flavonoid Content

The content of natural compounds, such as phenols and flavonoids were also determined through a colorimetric assay (Table 2). The total phenolic content (TPC) was highest in *V. herbacea*, whereas flavonoid concentration was most elevated in *V. major* var. *variegata*, followed by *V. herbacea*. Both *V. minor* and *V. major* displayed relatively low levels of phenols, as well as flavonoids. The calibration curves built for both phytochemical methods (Appendix A) had good linearity in the tested concentration interval. A good correlation (*r* = 0.8) was observed between the results obtained by HPLC and those from the colorimetric assay. Additionally, antioxidant activities had a high degree of correlation (*r* > 0.9) with TPC/TFC.

### 2.3. Determination of Total Alkaloid Content

The alkaloid content was determined by means of a colorimetric assay using UV-Vis spectrophotometry. Herein, the Dragendorff method with Munier and Macheboeuf modification (addition of tartaric acid in the reagent) was performed. There are two main differences compared to the original assay. First, the concentration of glacial acetic acid is higher, to determine indole alkaloids as well; second, the bismuth precipitate was not separated, but directly quantified for optical density at 560 nm (specific wavelength of the precipitate). Based on this working protocol, a calibration curve was built for vinblastine, with a degree of linearity high over 60 units of concentration (Figure 3a). TAC of all four plant extracts was determined using this method, with calculated values of vinblastine equivalents, as presented in Table 2. *V. minor*, followed by *V. herbacea*, had the highest alkaloid content. TAC obtained from Dragendorff assay was correlated with total alkaloid content from the HPLC results. HPLC results were calculated by addition of areas corresponding to all assigned alkaloids, based on the PCA analysis (Figure 3b). The value of the determined coefficient is very high (*r* = 0.999), indicating a strong correlation between the two methods, pointing out the ability of the used Dragendorff assay for the identification of indole alkaloids.

### 2.4. Antioxidant Activity

The antioxidant activity was determined for each diluted extract by three different assays: Rutin equivalent antioxidant capacity (REAC), cupric reducing antioxidant capacity (CUPRAC) and induced peroxidation of liposomes (LipPx). Throughout all three assays, the highest antioxidant activity was observed for *V. herbacea* extract, followed by *V. major* var. *variegata*, whereas the lowest one was that corresponding to *V. major*. Overall, the results of the applied assays correlated very well (for example, the correlation for REAC and CUPRAC was the highest (*r* > 0.9)). Surprisingly, despite the different underlying assay mechanisms (which could give opposite results, depending on the structure of the compounds tested), there was also a high degree of correlation between REAC, CUPRAC, and LipPx (*r* > 0.75). The actual values of rutin equivalents are showed in Table 2. Each one was calculated using the equation fitted for the calibration curve.

### 2.5. Antibacterial Activity and Morphological Examination

The agar well diffusion method revealed that all extracts had a better inhibitory effect against *S. aureus* compared to *E. coli* (Appendix A). The microdilution method confirmed the results from the agar well diffusion method, as growth of *S. aureus* was inhibited by all extract to a greater extent, compared to *E. coli* (*p* < 0.0001).

*E. coli* was inhibited by all extracts in a dose-dependent manner. The most efficient of the tested plant extracts was *V. minor*, which inhibited the bacterial growth at all tested concentrations (60% to 40% inhibition, Figure 4a). The inhibitory efficiency of *V. minor* extract was followed by *V. major*. On the other hand, *V. herbacea* and *V. major* var. *variegata* had a proliferative effect on *E. coli* at concentrations between 0.09–3.12%, as the corresponding optical densities exceeded by almost two-fold those of the untreated control (Figure 4c,d). The growth of *S. aureus* was inhibited in a dose-dependent manner by *V. minor* (Figure 4a) and *V. major* var. *variegata* extracts, with similar inhibitory capacities (Figure 4c). The *V. herbacea* extract inhibited the growth of *S. aureus* by 60%, at almost all concentrations (Figure 4d) and *V. major* (Figure 4b) was the least efficient extract.

A series of relations between alkaloid, phenolic contents and bacterial inhibition efficiency show that phenols are highly effective against *S. aureus*, but induce a proliferative response in *E. coli*, while the alkaloids can inhibit the *E. coli* growth, but not the *S. aureus* growth (Appendix A).

Scanning electron microscopy (SEM) analysis highlighted the morphological alterations in bacterial cells treated with *Vinca* extracts. *E. coli* treated with *V. major* var. *variegata* had a prominent glycocalyx present on the surface (Appendix A), as compared to the untreated control and the other extracts used. *S. aureus* reacted differently to the tested extracts (Figure 5). The surface of the cocci was smoother in the samples treated with *V. major* and *V. major* var. *variegata*, while *V. herbacea* had the most destructive effects, with visible signs of degradation.

### 2.6. Cytotoxicity Assays and Ultrastructural Examination

The cell viability was assessed in vitro through 3-(4,5-dimethylthiazol-2-yl)-2,5-diphenyltetrazolium bromide solution (MTT) assay on HaCaT keratinocytes and A375 melanoma (Figure 6).

According to ISSO: 10993-5, the viability is interpreted as following: Within 80–60%, weak cytotoxicity, 60–40%, moderate cytotoxicity, and <40%, strong cytotoxicity. All extracts reacted in a dose-depended manner and at concentrations below 1% (and including 2% for *V. minor*), the extracts had a proliferative effect on the HaCaT cells (viability >100%) but inhibited the A375 cells at all concentrations. 

The concentration at which 50% of the cells are inhibited (IC_50_) was calculated for each extract. These results enabled us to rank the toxicity of the extracts. Thus, for HaCaT cells, the *V. minor* extract was classified as least toxic (4.25%), followed by *V. major* (3.89%), *V. herbacea* (2.89%), and *V. major* var. *variegata* (2.85%). These values are in relation to the above-mentioned results, confirming that *V. minor* was the least toxic extract and *V. major* var. *variegata* the most toxic one (Figure 6a) against HaCaT cells. Overall, the viability of melanoma cells was affected to higher extents compared to normal keratinocytes (Figure 6b). However, the IC_50_ values were higher, indicating that higher doses of extracts are needed for cancer cell inhibition, along with longer exposure. *V. minor* (9%) was the least toxic extract against A375, as well (Figure 6b), followed by *V. major* (7.04%), *V. major* var. *variegata* (2.37%), and *V. herbacea* (2.3%).

Negative correlations (*p* > −0.6) were observed between IC_50_ values and TFC and TPC in both HaCaT and A375 cells (Appendix A). This indicates that once the concentration of phenols and flavonoids increase, the value of the IC_50_ concentration decreases, meaning that these phytoconstituents could be responsible for the cytotoxic effects on human cell lines.

Lactate dehydrogenase (LDH) release into the culture medium is an indicator for cell damage. Depending on the formulae used for calculation, the cell growth inhibition or necrosis could be determined [44]. For *V. minor* a dose-dependent reaction was observed on the HaCaT cells, while the rest of the extracts generated constant LDH values (Figure 7a). A375 cells were more affected compared to HaCaT cells (Figure 7c), when it comes to LDH release. The values exceeded even those observed for the Tween treated cells, for *V. herbacea*, followed by *V. minor*, but *V. major* and *V. major* var. *variegata* had also a pronounced negative effect against A375 cells. A negative correlation was observed between LDH release and TAC in HaCaT cells, and a positive correlation was observed in LDH release and TAC and TPC in A375 cells. This indicated that the alkaloids of *Vinca* species have a destructive effect on the membranes of melanoma cells but not normal cells (Appendix A).

High levels of nitric oxide (NO) can often lead to nitrite and nitrate oxidations, eventually inducing oxidative stress in cells [45]. Autoxidation of NO generates N_2_O_3_, a compound that reacts with sulfanilamide to produce a diazonium ion that is further coupled with *N*-(-1-napthyl)-ethylenediamine (N1-NAP), forming a product that strongly absorbs at 540 nm [45]. The Griess reaction was performed in order to determine the NO concentration in the culture medium, after the cells were individually treated with all extracts, for 48 h (HaCaT), and 72 h (A375), respectively. For HaCaT cells, a dose-dependent increase in NO concentration was observed for *V. minor* and *V. major* var. *variegata*. *V. herbacea* generated constantly high values of NO in the medium. Whereas, for *V. major* the values were constantly low, independent of the concentration used (Figure 7b). The NO concentration differed for A375, where high levels were detected for all *Vinca* extracts at 3% treatment (Figure 7d). A correlation (*p* = 8) was observed between the NO concentration and TPC and TAC in HaCaT cells that indicates how phenols and alkaloids of *Vinca* plant extracts influence the oxidation of nitrite and nitrate. Same correlation (*p* = 8) was observed for A375 cells and TAC (Appendix A).

To further explore the effects of the plant extracts, transmission electron microscopy (TEM) analysis was employed to reveal ultrastructural changes in the treated cells. A375 cells treated with *V. minor* extract showed numerous vesicles, while those treated with the rest of the extracts presented a high number of mitochondria (Figure 8). Similar results were observed on the HaCaT cells (Appendix A).

## 3. Discussion

Hydroalcoholic extracts were obtained from the leaves of *Vinca* species, which are known as medicinal plants and having a high content of alkaloids, flavonoids, and phenolic compounds, and well-documented antibacterial, antioxidant, and cytotoxic effects [46,47,48]. The chemical composition of plant extracts is dependent on the plant species, organ and tissue used, harvesting period, environmental conditions, extraction method, and solvent used for preparation [33,49,50]. Therefore, the leaves of *Vinca* species are preferred for extract preparation due to the highest alkaloid content stored in a short period [20,51]. For example, in *Catharanthus roseus* one of the well-known plant of the Apocynaceae family, some of the biosynthetic genes of vinblastine are expressed in epidermal cells of the leaves of the plant [52]. Moreover, since the crude extracts of the species considered herein are an important source of phytoconstituents, the TAC, TPC, and TFC were determined for all four extracts (Table 1 and Table 2).

One objective was to determine the total alkaloid content by a fast and reliable assay, preferably using a relatively cheap and accessible experimental setup. A well-known colorimetric method is Dragendorff assay. This relies on the following principle: In acidic environment, Bi^3+^ forms a complex anion with I^−^, [BiI_4_]^−^, which in turn, can interact with the positive charge on the nitrogen-containing alkaloids, resulting in a fine orange precipitate. Unfortunately, not all alkaloids display a quaternary nitrogen atom at regular pH, being the case of indole alkaloids. These representatives contain only a tertiary nitrogen atom, without charge, making their detection by Dragendorff reagent very difficult. Still, as shown here, if the optimum quantity of acid is added, the indole alkaloids can be protonated, thus, precipitating in the presence of the [BiI_4_]^−^.

The *Vinca* extracts analyzed by HPLC have terpene indole alkaloids. This implies that the plant species share the metabolic pathways of biosynthesis from tryptophan and secologanin precursors [30,52]. This could indicate that the structure of the unidentified terpene indole alkaloids is very similar to that of vincamine, explaining the reason why all extracts, except for *V. herbacea*, had traces of it [21,32,53]. *V. minor* had the highest amount of vincamine (65 µg/g) and compared to other studies that reported a content of 0.057% [33] or 78.9 ng/g [54]. The method used, herein, was proved very efficient for alkaloid extraction.

According to the PCA results, there are many other alkaloids present in the studied extracts, some of them possibly being indole type as well. However, the relatively low concentration of alkaloids is not surprising. Their synthesis is strictly regulated, as they can have detrimental effects on the plant itself. Besides the alkaloids, three other main phytoconstituents classes are present: Flavonoids, hydroxybenzoic and hydroxycinnamic acids, which are known to be involved in plant protection against various abiotic or biotic forms of stress [55].

All four *Vinca* extracts exhibited remarkably high levels of chlorogenic acid, a natural compound synthesized via the phenylpropanoid pathway from phenylalanine, under various forms of stress [56]. This result can partly be explained by the interplay identified between chlorogenic acids and alkaloids in other plants, especially when a biotic stress is applied. For example, interesting relationships were described between chlorogenic acids and pyrrolizidine alkaloids: Depending on the bioavailable form of the alkaloid, there can be synergistic (if the alkaloid is an oxide) or antagonistic effects manifested [57]. Other natural compounds were identified, such as rutin, quercetin, and caffeic acid. Moreover, HPLC revealed additional unidentified peaks that were attributed to unknown, possibly new, compounds belonging to these classes.

Phytoconstituents in *Vinca* extracts are species-dependent and have synergistic action. They are responsible for the antioxidant activity (Table 2), antibacterial (Figure 4) and cytotoxic effects (Figure 5), which were highly correlated. The results showed that *V. herbacea* extract had the highest antioxidant activity, followed by *V. major* var. *variegata*, whereas the lowest one was attributed to *V. major*. The antioxidant potential of plant extracts is determined by different phytoconstituents, such as vincamine [58], chlorogenic acid [42], and other phenolics [59]. These findings confirm that the antioxidant potential is different for *V. minor* [60], *V. herbacea* [61], and *V. major* [16].

The antibacterial activity was species- and dose-dependent. *V. minor* had the best inhibitory effect against both *S. aureus* and *E. coli*, which is most probably generated by the phytoconstituents. Similar results were obtained with *V. minor* extracts against Gram-positive bacteria and a lower inhibitory capacity against the Gram-negative ones [60]. These inhibitory effects are determined by the antibacterial compounds present, such as vincamine [62] and common plant flavonoids such as rutin, chlorogenic acid, caffeic acids, isoquercitrin, quercitrin, quercetin, etc. [63], that act synergistically. Another report proved that vincamine is effective against *E. coli* and *S*. *aureus* [62], and that the flavonoids rutin, chlorogenic acid, quercetin, or kaempferol were generally more effective against *E. coli* and *Pseudomonas aeruginosa* (Gram-negative), than against *Enterococcus fecalis* and *S. aureus* (Gram-positive) [63]. The *V. herbacea* extract had a proliferative effect on *E. coli* and an inhibitory effect on *S. aureus*. Also, this extract contained elevated levels of rutin, a compound with good antibacterial potential [8]. Different antibacterial effects are mentioned for rutin, with a greater inhibitory activity determined against *S. aureus* compared to *E. coli* [64], but also against *Aeromonas hydrophyla* [5]. This also applies to *V. major* and *V. major* var. *variegata*, even if they had weaker inhibitory potential against the tested bacterial strains. However, the potential for these extracts is still to be uncovered in further analyses, where higher doses or even their combinations could be tested.

The morphological alterations, generated by the plant extracts on the bacterial cells, were also assessed through SEM for a better understanding of the occurring processes. The *E. coli* bacilli were not affected significantly compared to the untreated control (Appendix A). However, a proliferation of the extracellular matrix (glycocalyx and flagella) was observed on the samples treated with *V. major* var. *variegata* and *V. herbacea* and this could explain the proliferative effect observed in the microdilution method [65,66]. The *S. aureus* cocci were found in different stages of degradation when treated with the *Vinca* extracts (Figure 5), and the ones treated with *V. herbacea* showed signs of membrane disruption, similar to other findings [67]. Still, the mode of action of phenols and alkaloids against bacteria is yet to be elucidated, so that the base for new and alternative antibiotics could be settled [1].

The *Vinca* extracts had rather proliferative effects on HaCaT cells at concentrations between 0.09–2%, and inhibitory effects for the 3% concentration used. The obtained results could be explained by a dose-dependent cytotoxic effect on the cell lines [68]. The melanoma cells (A375) were negatively affected by all extracts in a dose-dependent manner as well, but *V. herbacea* and *V. major* var. *variegata* showed also a promising anticancer activity. The mechanism of action of *Vinca* alkaloids (vinblastine and vincristine) [69] is by interaction with β-tubulin at a region adjacent to the GDP-binding site, at the plus end of microtubules, known as the ‘vinca domain’. These have different modes of actions (inhibition or acceleration of cell division), depending on the concentration [70]. Consequently, the vinca domain could bind other *Vinca* alkaloids, only this time the effects could be the opposite of apoptosis as we observed. However, to our knowledge, there is no literature available to confirm or infirm this theory.

Additionally, the ultrastructural changes induced by the *Vinca* extracts on the human cell lines, were assessed thorough TEM. As expected, the HaCaT cells which expressed high viability also had a high number of mitochondria, compared to untreated controls (Appendix A), and this could be due to the phenolic content present in the plant extracts [71]. On the other hand, the A375 cells were most affected by the *V. minor* extract as indicated by the ultrastructural analysis (Figure 8), where multiple lysosomes were detected. This is a clear sign of necrosis, a premature death of the cells caused by the extract [72], and similar results were previously reported [73].

Henceforth, more complex analyses are required to determine the mode of action against pathogens and the anticancer potential of *Vinca* alkaloids, while the synergistic action of *Vinca* extracts will be exploited in a future research.

## 4. Materials and Methods

### 4.1. Plant Material and Extract Preparation

*Vinca minor* L. (lesser periwinkle), *Vinca major* L. (great periwinkle), *Vinca major* L. var. *variegata* ‘Louden’, and *V. herbacea* Waldst. and Kit. (herbaceous periwinkle) were collected from ‘Alexandru Borza’ Botanical Garden of Cluj-Napoca, Romania, and identified by Pârvu Marcel. A voucher specimen for each species was deposited in the Herbarium of ’Babeş-Bolyai’ University, Cluj-Napoca, Romania (*V. minor* CL 665977, *V. major* CL 668019, *V. major* var. *variegata* CL 668018, *V. herbacea* CL 668021).

During the flowering season (April–May and September), the fully developed leaves of adult plants were collected and thoroughly washed with tap and distilled water, cut into small fragments (0.5–1 cm), weighed, and placed in the percolator. The extraction of phytoconstituents was conducted as previously described [9], through the cold repercolation method in a 1:2 (*w:v*) solvent to fresh herba ratio. For three days the fresh herba was extracted with 70% ethanol (Merck, Bucharest, Romania), at room temperature. The fluid extracts obtained by filtration were as follows (*w:v*/g:mL): 1:1.2 (*V. minor*), 1:1.4 (*V. major*), 1:1.5 (*V. major* var. *variegata*), and 1:2 (*V. herbacea*). The final ethanol concentration was 30% in all obtained extracts.

### 4.2. Phytochemical Analyses of the Vinca Leaf Extracts

#### 4.2.1. HPLC-DAD Method

All reagents were of analytical grade purity and unless otherwise specified. All reagents were acquired from Sigma-Aldrich (Merck, Bucharest, Romania).

The chromatographic separations and detection were performed as previously described [74], on an Agilent 1200 HPLC system (Agilent Technologies Inc., Waldbronn, Germany) equipped with a vacuum degasser and temperature-controlled sample tray. A quaternary pump controlled the mobile phase flow, and the samples were automatically injected. The chromatographic separations were run on a Zorbax SB-C18 column (250 mm × 4.6 mm, 5 μm particle size) from Agilent (Agilent Technologies Inc., Santa Clara, CA, USA), placed in a column thermostat compartment, and the detection was accomplished via a DAD detector. The injection volume was 8 μL (0.22 μm filtered extract), the column temperature was set to 30 °C, and the flow rate was 1 mL/min.

The finally employed optimum method consisted of a multistep gradient elution system using 0.1% trifluoroacetic acid in ultrapure water as solvent A and acetonitrile as solvent B. The steps of the gradient were as follows: 0–2 min isocratic at 8% B, 2–17 min from 8 to 30% B, 17–27 min isocratic at 30% B, 27–37 min from 30% to 85% B, 37–40 min from 85% to 95% B, 40–41 min isocratic at 95% B, and 41–41.1 min back to 8% B, where was kept until min 44. The UV-Vis detection of the compounds was performed using the DAD detector that measured the entire spectrum in 210–600 nm region (2 nm resolution), every 2 s; the chromatograms were monitored at 242, 260, 280, 320, and 340 nm. The standards used were: 3,4-dihydroxybenzoic acid, chlorogenic acid, 4-hydroxybenzoic acid, caffeic acid, syringic acid, rutin, *p*-coumaric acid, isoquercitrin, ferulic acid, quercitrin, myricetin, berbamine, vincamine, jatrorrhizine, quercetin, palmatine, berberine, kaempferol, vinblastine, and galangin. Calibration curve was constructed using a mixture of the above-mentioned standards at 35, 53, 70, 105, 140, 210, 280 μg/mL, and the area of the peak by integration employed by the Agilent soft. Identification of the compounds from the analyzed samples was done using both chromatographic retention time and UV-Vis spectral similarities that were done by the built-in soft with the spectra of the analytical standards. The chemometric investigations of the spectral features of chromatographic peaks that contribute to at least 1% of the total peak area at 230 nm, were done as previously described [75]; before performing PCA analysis, all chromatograms were normalized by min-max method.

#### 4.2.2. Alkaloid Content

The total alkaloid content (TAC) was determined by an adapted Dragendorff assay [76]. The first solution consisted of 43.8 mM Bi(NO_3_)_3_ and 1.66 M tartaric acid, in 20% acetic acid (aqueous solution). A second solution was obtained by dissolving KI in ultrapure water to a final concentration of 2.4 M. Both solutions were shaken vigorously until homogenization. Afterwards, 5 mL of the first solution were mixed with 2 mL of the second solution. Next, 260 μL of this reagent were mixed thoroughly with 30 μL of 100 μg/mL of each extract, and 10 μL of glacial acetic acid. The optical density was registered using a Tecan multiplate reader (Tecan Trading AG, Mannedorf, Switzerland) at 560 nm. The standard used for the calibration curve was vinblastine (concentration range: 1 to 20 μg/mL) and all measurements were performed in duplicate.

#### 4.2.3. Colorimetric Determination of Phytoconstituents in the *Vinca* Leaf Extracts

For the determination assays of all phytoconstituent classes, the extracts were diluted to a concentration of 100 mg/mL in 30% ethanol solution. The total phenolic content (TPC) was determined by Folin-Ciocâlteu reducing capacity assay [77]: 25 μL of each extract were mixed with 25 μL of Folin-Ciocâlteu reagent and 200 μL of ultrapure water, and incubated for 5 min. Then, 25 μL of Na_2_CO_3_ solution (stock concentration: 10.6 g/100 mL) were added and the resulting mixture was further incubated for 60 min in the dark. The absorbance was measured at 725 nm before, and after, the 60 min incubation period. The corresponding percentages were calculated as described in the literature. The standard used for the calibration curve was gallic acid (in 2–40 μg/mL range).

The total flavonoid content (TFC) was determined by AlCl_3_ complexation (procedure 1) [78]. Thus, 10 μL of each extract were mixed with 50 µL of 2% AlCl3 solution and 50 µL of 1 M sodium acetate in 140 μL ultrapure water, in a 96-well plate. After an incubation period of about 5 min, 50 µL of 0.1 mM HCl were added under vigorous shaking. After an additional 20 min, the absorbance was measured at 452 nm using a Tecan Spark multiplate reader. The standard used for the calibration curve was rutin (in 1.6–50 μg/mL range). All the above assays were performed in duplicates.

### 4.3. In Vitro Antioxidant Activity

All reagents were of analytical grade purity, and unless specified otherwise, all reagents were acquired from Sigma-Aldrich (Merck, Bucharest, Romania).

For the antioxidant capacity assays performed, the extracts were diluted to a concentration of 100 mg/mL in 30% ethanol solution. Rutin Equivalent Antioxidant Capacity (REAC), in fact ABTS bleaching assay with rutin as standard [79] and cupric ion reducing antioxidant capacity (CUPRAC) [79] are presented in the Appendix A.

Induced peroxidation of liposomes (LipPx) assay was performed as follows: Liposomes of 0.2 mg/mL concentration in phosphate buffer saline (PBS) of 7.4 pH were sonicated for 30 min until homogenization [80]. Further, 290 μL of this solution was mixed with 10 μL of each extract (10 μg/mL) and the reaction was triggered by the addition of cytochrome C, to a final concentration of 2 μM. The reaction evolution was recorded at 235 nm using the Tecan Spark multiplate reader overnight, at 25 °C. The corresponding lag phase was determined as the value of inflexion point for all kinetic measurements. Rutin (in 0.067–2.3 μg/mL range) was used as a standard to obtain the kinetic curve.

### 4.4. Antibacterial Activities

The antibacterial activity of the plant extracts was tested on a Gram-positive model, *Staphylococcus aureus* (ATCC: 25923) and a Gram-negative model, *E. coli* (ATCC: 25922), using the agar-well diffusion and microdilution methods adapted after Carpa, et al. [81]. The bacterial strains were incubated at 37 °C, for 24 h, in Petri dishes with Mueller-Hinton (MH) agar media (VWR Chemicals, VWR International, Darmstadt, Germany), and were further adjusted to a 0.5 McFarland turbidity standard, according to EUCAST protocols [82].

#### 4.4.1. Agar-Well Diffusion Method

For the agar-well diffusion method Petri dishes with MH-agar media were inoculated with each bacterial strain and left at room temperature for 30 min to infiltrate. Subsequently, 6 mm diameter wells were carved in the agar using a cut sterile pipette tip. The wells were then filled with sterile cotton beads. Each bead was loaded with 150 µL of each extract. The plate also contained a vehicle control with 30% ethanol and a positive control with ciprofloxacin (CIP, 5 µg/mL concentration). After 24 h incubation at 37 °C, the formed hallow around the wells was measured. Each experiment was conducted six times and the mean was calculated.

#### 4.4.2. Microdilution Method

For the microdilution method, 96-well plates were prepared as following: 100 µL MH-broth media in each well and 100 µL of each extract was inoculated (in the first row only), and two-fold serial dilutions were performed, with concentrations ranging from 50% to 0.09%. All wells were then filled with 10 µL of bacterial suspensions, with final concentrations of 5 × 10^5^ CFU/mL (colony forming units). Each plate had an untreated control and a positive control with 4.5% CIP. Vehicle controls with ethanol and media with extracts were also prepared in separate plates. The plates were incubated for 24 h at 37 °C and the absorbance was read at 600 nm, using the BioTech Epoch plate reader (BioTek Instruments, Winooski, VT, USA) and Gen5 Software (version 1.09).

Bacterial inhibition was expressed as inhibition percentages and calculated according to Equation (1):(1)$$\mathrm{Inhibition}\;\left(\%\right)\;=\;\frac{\mathrm{registered}\;\mathrm{absorbance}\;(\mathrm{nm})}{\mathrm{untreated}\;\mathrm{control}\;(\mathrm{nm})}\;100$$

#### 4.4.3. Morphological Examination through Scanning Electron Microscopy

Scanning electron microscopy (SEM) was used to assess the possible morphological damages induced on the bacterial strains. The method was adapted after an existing protocol [83] and modified as follows; the fixator solution consisted of 0.5% alcian blue, 2% glutaraldehyde, and 2% paraformaldehyde in 0.15 M PBS (Merck, Bucharest, Romania); the bacteria were incubated for 7 h at 4 °C; dehydration occurred in acetone in different concentrations (30%, 50%, 70%, 80%, 90%, and 100%), for 10 min each step, at 4 °C. Additionally, the samples were post-fixed with hexamethyldisilazane (1:1 with acetone, and 1:0, 10 min). The samples were examined with SEM Hitachi SU8230 (Hitachi, Tokyo, Japan).

### 4.5. In Vitro Cytotoxicity Assays

#### 4.5.1. Biochemical Assays

Cytotoxicity assays were conducted on normal human keratinocytes (HaCaT 300493, CLS, Heidelberg, Germany) and skin melanoma (A375, ATCC CRL-1619), as previously described [84]. Briefly, the cells were cultured on 25 cm^2^ plastic dishes in Dulbecco’s Modified Eagle’s media (DMEM) with 4.5 g/L glucose (Lonza Group Ltd., Basel, Switzerland), supplemented with 10% fetal calf serum (FCS; Ghibco, Thermo Fisher Scientific, Paisley, UK), 1% penicillin-streptomycin and 1% l-glutamine (Lonza Group Ltd., Basel, Switzerland). The plates were kept in a humidified incubator at 37 °C with 5% CO_2_ atmosphere. At 80% confluence, the cells were harvested using 0.25% trypsin-EDTA (Lonza Group Ltd., Basel, Switzerland) and sub-cultured in 96-well plates.

Aliquots of 100 µL of the prepared cells were plated in each well, at a density of 12 × 10^3^ cells/well (HaCaT) and 10^4^ cells/well (A375) and were left to attach for 24 h. After incubation, the culture media was replaced with media containing plant extracts in final concentrations ranging from 0.09% to 3%, and each plate had untreated, negative, and positive controls, depending on the assay (cells with 2% Tween 20 solution and lipopolysaccharides). Based on preliminary investigations and on the consulted literature, the HaCaT cells were left with the plant extracts for 48 h and the A375 cells, for 72 h. The viability was analyzed through MTT assay, membrane integrity was evaluated based on the LDH assay, and signs of oxidative stress were determined through the NO assay. The detailed protocols are presented in the Appendix A.

Additionally, the median inhibitory concentration (IC_50_) was calculated according to Equation (2):(2)$$x\;=\;\frac{y\;-\;c}{m}$$ where x is the median concentration, y is 50%, m is the coefficient calculated from the exponential fit of the data, and c is a constant generated by the exponential fitting of the data.

#### 4.5.2. Ultrastructural Investigation through Transmission Electron Microscopy

The ultrastructural modifications were investigated through transmission electron microscopy (TEM). The samples were prepared as previously described [44] and adapted after Hayat [85]. The detailed protocol is presented in the Appendix A. The samples were examined using TEM Jeol JEM 1010 (Jeol, Tokyo, Japan) and SEM Hitachi SU8230.

### 4.6. Statistical Analyses

Each phytochemical analysis and antioxidant activity was performed in duplicate, and the mean and standard deviation were then calculated. In the biological analyses, each of the concentrations for the cytotoxic and antibacterial activities was tested six times, and all data refer to the mean ± standard error of at least four independent experiments. The statistical analyses performed were: Principal Component Analysis (performed in STATISTICA 12), Pearson’s correlation coefficient (*r*), Spearman’s rank correlation coefficient (ρ), Student’s t-test, and one-way ANOVA statistical analyses performed using Origin 8 software (Origin Lab Corporation, Northampton, MA, USA). Values of *p* ≤ 0.05 were considered statistically significant.

## 5. Conclusions

The chemical composition of *Vinca minor*, *V. major*, *V. major* var. *variegata*, and *V. herbacea* is correlated with the observed pharmacological activities, such as antioxidant, antibacterial, and cytotoxic potentials, which brings novelty to the field. The total alkaloid content was determined for the first time in these hydroalcoholic extracts, revealing that *V. minor* was the richest in alkaloids and that traces of vincamine are present in *V. major* and *V. major* var. *variegata*. Surprisingly, *V. herbacea* had a remarkably high rutin content. Based on this complex chemical composition, the antioxidant activity, antibacterial potential, and cytotoxic effects were species- and dose-dependent. The high antioxidant potential of *V. herbacea* could be related to the high amount of rutin, and because *V. minor* had the highest alkaloid content, the antibacterial effects were greater for this extract. Of course, the generated effects were due to the synergistic action of the plethora of phytoconstituents, and this was also reflected in the cytotoxic activity against HaCaT and A375 cellular lines. Under the current circumstances, all tested plants kept their pharmacological potential. Moreover, *V. herbacea* and *V. major* var. *variegata* acted similar to *V. minor*. The chemical profile of the *Vinca* plant extracts, together with the observed effects in the treated cells, holds valuable information for the development of new alternative treatments. Still, the mode of action of distinctly detected compounds must be characterized, and unknown compounds are yet to be identified.

## Figures and Tables

**Figure 1 molecules-26-03040-f001:**
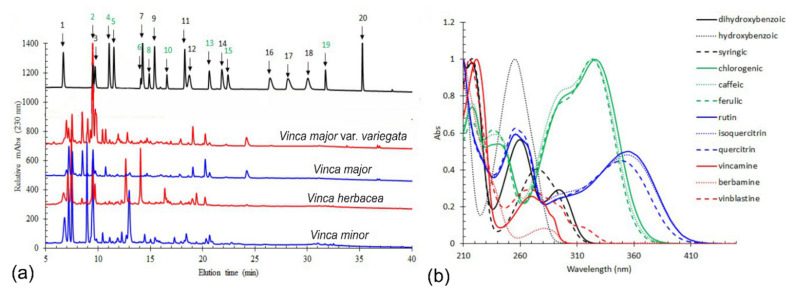
(**a**) HPLC-DAD chromatograms of the *Vinca minor*, *V. major*, *V. major* var. *variegata*, and *V. herbacea* plant extracts monitored at 230 nm. Chromatogram of analytical standards includes: (1) 3,4-dihydroxybenzoic acid, (2) chlorogenic acid, (3) 4-hydroxybenzoic acid, (4) caffeic acid, (5) syringic acid, (6) rutin, (7) *p*-coumaric acid, (8) isoquercitrin, (9) ferulic acid, (10) quercitrin, (11) myricetin, (12) berbamine, (13) vincamine, (14) jatrorrhizine, (15) quercetin, (16) palmatine, (17) berberine, (18) kaempferol, (19) vinblastine, and (20) galangin. The identified compounds in the studied *Vinca* species are marked in green in the upper chromatogram; (**b**) UV molecular absorption spectra as registered by the DAD detector for the representative standards from each main phytochemical group (grey-hydroxybenzoic acids, green-cinnamic acids, blue-flavonoids, and red-alkaloids).

**Figure 2 molecules-26-03040-f002:**
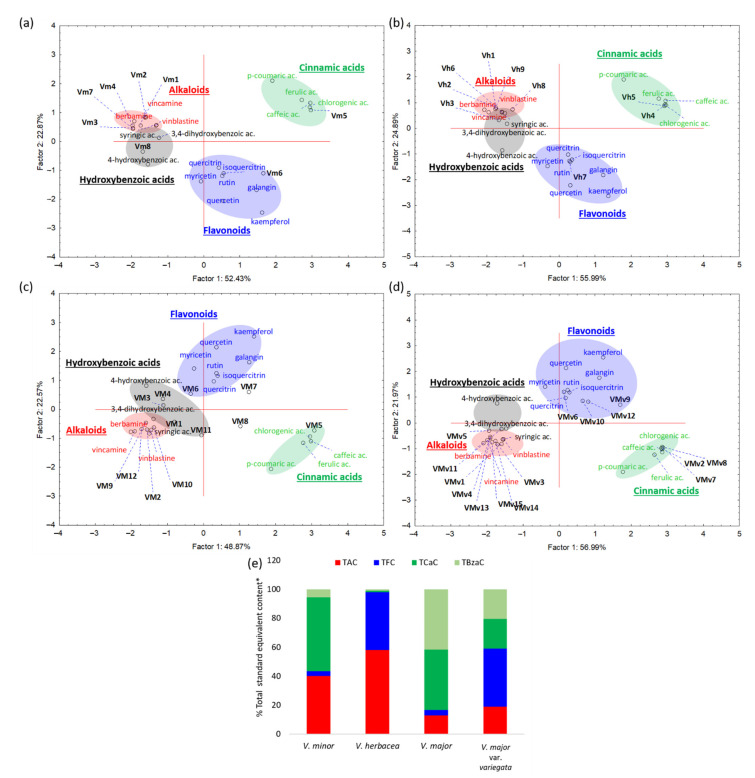
Chemo-mapping of the major chromatographic peaks—phytoconstituents classification—based on spectral similarities for each studied extract using PCA. PCA was applied on exported UV DAD spectra, for each chromatographic peak. Shown here are the scatterplots of the scores for the first two principal components for (**a**) *V. minor*, (**b**) *V. herbacea*, (**c**) *V. major*, and (**d**) *V. major* var. *variegata*. The classified compounds are detailed in Appendix A. Groups with high similarity are clustered in specific color for each phytoconstituent group or class (grey-hydroxybenzoic acids, green-cinnamic acids, blue-flavonoids, and red-alkaloids); (**e**) Total standard (vincamine for TAC, quercetin for TFC, cynnamic acid for TCaC and benzoic acid for TBzaC) equivalent content (%) for each extract for 230 nm chromatogram, after PCA classification. TAC-total alkaloid content, TFC-total flavonoid content, TCaC-total cinnamic acids content, TBzaC-total hydroxybenzoic acids content, Vm = *V. minor*, VM = *V. major*, VMv = *V. major* var. *variegata*, Vh = *V. herbacea*.

**Figure 3 molecules-26-03040-f003:**
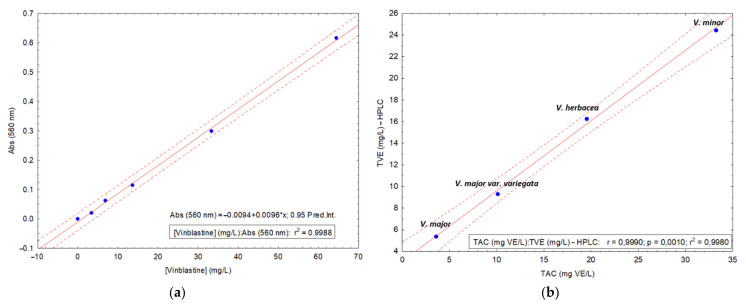
(**a**) Calibration curve of the proposed Dragendorff assay with vinblastine as a standard; (**b**) graphical correlation between Dragendorff TAC and HPLC calculated Total Vincamine Equivalents.

**Figure 4 molecules-26-03040-f004:**
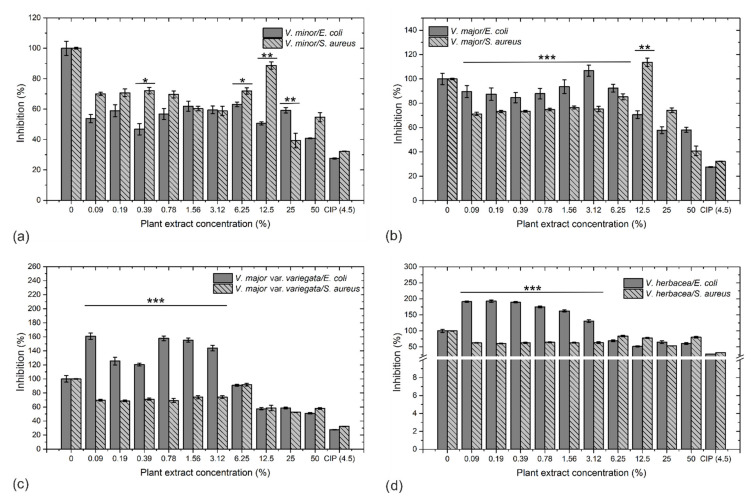
The antibacterial effect of *Vinca minor* (**a**), *V. major* (**b**), *V. major* var. *variegata* (**c**), and *V. herbacea* (**d**) leaf extracts against *E. coli* and *S. aureus*, assessed through the microdilutions method. The values represent the mean of at least three independent experiments ± standard error of the mean (s.e.m.); *** *p* < 0.0001, ** *p* < 0.001, * *p* < 0.05 according to one way ANOVA and Student’s *t* tests.

**Figure 5 molecules-26-03040-f005:**
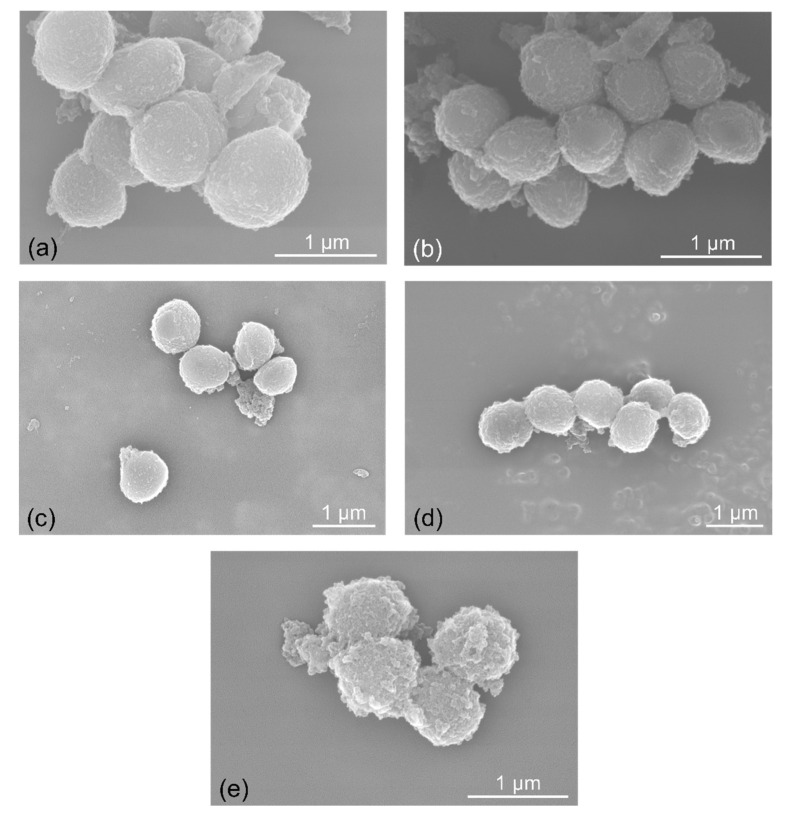
SEM micrographs of the untreated *S. aureus* strain, (**a**) and compared to the strains treated with (**b**) *V. minor*, (**c**) *V. major*, (**d**) *V. major* var. *variegata*, and (**e**) *V. herbacea* leaf extracts at Minimal Inhibitory Concentration (MIC).

**Figure 6 molecules-26-03040-f006:**
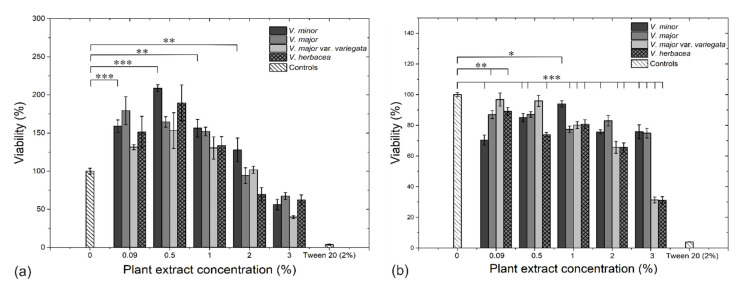
In vitro cytotoxic effects of the *Vinca* plant extracts used against normal keratinocytes (HaCaT) for 48 h (**a**), and skin melanoma cells (A375) at 72 h (**b**), compared to a positive control (untreated cells) and a negative control (cells treated with Tween 20 at 2% concentration); *** *p* < 0.0001, ** *p* < 0.001, * *p* < 0.05 according to one way ANOVA and Student’s *t* tests.

**Figure 7 molecules-26-03040-f007:**
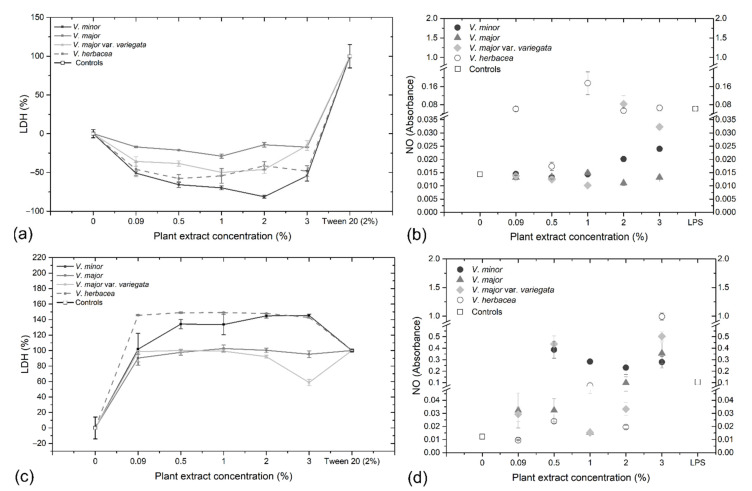
In vitro analysis of LDH (**a**) and NO (**b**) release in normal keratinocytes (HaCaT) treated with the plant extracts for 48 h and in vitro analysis of LDH (**c**) and NO (**d**) release in skin melanoma cells (A375) treated with the *Vinca* leaf extracts for 72 h; LPS = lipopolysaccharides.

**Figure 8 molecules-26-03040-f008:**
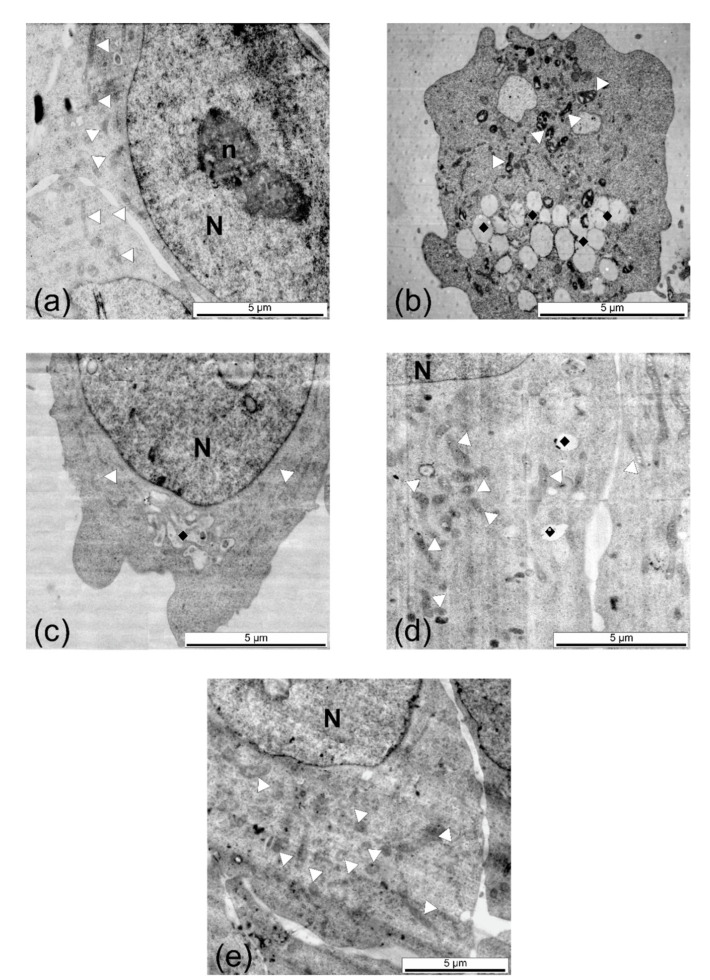
TEM micrographs of untreated A375 melanoma cells (**a**) and compared to the cells treated with (**b**) *V. minor*, (**c**) *V. major*, (**d**) *V. major* var. *variegata*, and (**e**) *V. herbacea* leaf extracts at IC_50_ values. White triangle = mitochondria, black rhombs = vesicles/lysosomes, N = nucleus, n = nucleolus.

**Table 1 molecules-26-03040-t001:** Concentration of the main phytoconstituents identified in the *Vinca minor*, *V. major*, *V. major* var. *variegata*, and *V. herbacea* leaf extracts as determined and calculated from HPLC chromatograms.

Compound	Compound Concentration Expresses in μg/g			
	*V. minor*	*V. major*	*V. major* var. *variegata*	*V. herbacea*
Chlorogenic acid	4112 ± 13 **	675 ± 160	932 ± 260	1538 ± 200 *
Caffeic acid	229 ± 2 ***	13 ± 2	182 ± 23 **	13 ± 1
Rutin	73 ± 1 **	11 ± 1	94 ± 10 **	2528 ± 160 ***
Isoquercitrin	12 ± 1	12 ± 1	38 ± 4 *	87 ± 4 ***
Quercitrin	52 ± 4 *	11 ± 1	45 ± 8 *	109 ± 10 ***
Vincamine	65 ± 1 ***	42 ± 3	31 ± 2	n.d.
Quercetin	21 ± 2	14 ± 2	28 ± 2	20 ± 3

n.d.—not detected; *** *p* < 0.001, ** *p* < 0.01, * *p* < 0.05 according to Student’s *t* tests.

**Table 2 molecules-26-03040-t002:** Antioxidant activities and phytoconstituent content of *Vinca minor*, *V. major*, *V. major* var. *variegata*, and *V. herbacea* leaf extracts expressed as equivalents of standard for each assay.

Extract	REAC	CUPRAC	LipPx	TPC	TFC	TAC
	mg rutin/g	mg rutin/g	mg rutin/g	mg gallic/g	mg rutin/g	mg vinblastine/g
*V. minor*	106 ± 53 *	123 ± 18 *	43 ± 9	51 ± 0 *	29 ± 1	332 ± 33 ***
*V. major*	35 ± 13	50 ± 2	153 ± 12 **	21 ± 2	11 ± 3	35 ± 11
*V. major* var. *variegata*	335 ± 10 **	463 ± 37 **	229 ± 20 **	93 ± 7 **	179 ± 2 ***	101 ± 6 *
*V. herbacea*	405 ± 29 **	583 ± 48 **	267 ± 23 **	100 ± 6 **	151 ± 1 ***	195 ± 9 **

REAC = rutin equivalent antioxidant capacity, CUPRAC = cupric reducing antioxidant capacity, LipPx = inhibition of induced liposomes peroxidation, TPC = total phenolic content, TFC = total flavonoid content, TAC = total alkaloid content; *** *p* < 0.001, ** *p* < 0.01; * *p* < 0.05 according to Student’s *t* test.

## Data Availability

Samples of the data are provided by the corresponding author on request.

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
