# Peer review of "The Phytochemical Analysis of *Vinca* L. Species Leaf Extracts Is Correlated with the Antioxidant, Antibacterial, and Antitumor Effects"

_molecules, 2021, doi:10.3390/molecules26103040_

Round 1
Reviewer 1 Report
The manuscript entitled “The phytochemical analysis of Vinca L. species leaf extracts is correlated with the antioxidant, antibacterial, and antitumor effects” is aimed at characterizing and comparing the chemical composition of the leaves extracts of several Vinca species. Searching for correlation between several groups of constituents and the antibacterial activity against E. coli and S. aureus, and cytotoxic effects against normal keratinocytes and skin melanoma human cell lines was a further goal of the research. A novel approach for the determination of the total content of alkaloids was proposed; finally, SEM and TEM techniques were applied to study ultrastructural modifications induced in human cell lines by the extracts.
The paper is well-written and focuses on an interesting topic. A lot of data are presented, and the discussion is interesting. However, some minor and major points have to be clarified or improved. They are listed below
Introduction
The introduction is well written and quite well focused. Two little question/suggestions:
- Lines 60-66: in these lines, the authors focus the attention on the vincamine. Is it the only interesting molecule among the ones identified in the Vinca species so far?
- Please, remove the sentence in lines 84-86. It does not provide any scientific information, and the reader itself will decide if “the information provided herein is necessary and brings novelty in 85 the literature”
Materials and methods
- Lines 419-420: Please, provide more information about the extraction process, as for example the drug/solvent ratio, if there was a sort of agitation and so on.
- Line 442: what “the second protocol” refers to?
- Lines 442-445: how these standards have been selected? Are they molecules already reported as present in some Vinca species? Please, specify. Some lines must be added in the introduction section, with more information about the types of molecules identified to date in the vinca species.
- In miss in the M&M section the description of the rutin equivalent antioxidant capacity (REAC).
- Lines 488-495: do this part describe the “Induced peroxidation of liposomes (LipPx)”? Please specify
Results
- Figure 1, Figure caption: “the identified compounds are marked in green”. Do authors mean the compounds identified in the Vinca samples? Please, specify
- Table 1: vinblastine was not detected in any of the vinca species. Why is it listed in the table? Since the other not detected compounds are not listed in the table, it should be removed.
- Tables and figures captions and footnotes. What type of statistical analysis the symbols “*”, “**”, “***” refer to? More in general, please add in the material and method section a paragraph with detailed information about the performed statistical analysis.
- Lines 131ff: in the material and methods sections, authors state that, starting from 20 commercial standard, they built calibration lines. Consequently, I would have expected to find in the manuscript quantitative data about the identified molecules and not the sums of areas of peaks related to each category of compounds. Even for those peaks not attributed to a specific molecule, but included in one of the four categories (flavonoids, cinnamic acids, hydroxybenzoic acid, alkaloids), a quantitative datum can be easily calculated using the calibration curve of a standard belonging to that category. This part has to be definitely improved, and the other part of the manuscript, in which quantitative data from HPLC analysis have been used for example for identifying correlations, have to been changed accordingly.
Based on these point, I recommend major revision
Author Response
Response to Reviewer’s 1 indications
Q1. The manuscript entitled “The phytochemical analysis of Vinca L. species leaf extracts is correlated with the antioxidant, antibacterial, and antitumor effects” is aimed at characterizing and comparing the chemical composition of the leaves extracts of several Vinca species. Searching for correlation between several groups of constituents and the antibacterial activity against E. coli and S. aureus, and cytotoxic effects against normal keratinocytes and skin melanoma human cell lines was a further goal of the research. A novel approach for the determination of the total content of alkaloids was proposed; finally, SEM and TEM techniques were applied to study ultrastructural modifications induced in human cell lines by the extracts. The paper is well-written and focuses on an interesting topic. A lot of data are presented, and the discussion is interesting. However, some minor and major points have to be clarified or improved.
A1. We would like to thank the reviewer for their support and given indications. The responses are written point by point bellow and are marked in yellow in the main text of the manuscript.
Q2. Lines 60-66: in these lines, the authors focus the attention on the vincamine. Is it the only interesting molecule among the ones identified in the Vinca species so far?
A2. Additional information was added along with supplementary references from line 60 to 72 and from 77 to 83.
Q3. Please, remove the sentence in lines 84-86. It does not provide any scientific information, and the reader itself will decide if “the information provided herein is necessary and brings novelty in 85 the literature”.
A3. The sentence was removed.
Q4. Lines 419-420: Please, provide more information about the extraction process, as for example the drug/solvent ratio, if there was a sort of agitation and so on.
A4. Thank you, the missing information was added.
Q5. Line 442: what “the second protocol” refers to?
A5. There we made a mistake in trying to express the fact that whereas the whole spectrum was monitored between 210 and 600 nm, and that the chromatograms were only recorded at some specific wavelengths to allow the identification of targeted compounds. We have modified the sentence in M&M section accordingly.
Q6. Lines 442-445: how these standards have been selected? Are they molecules already reported as present in some Vinca species? Please, specify.
A6. Polyphenolic standards had been selected after a literature study concerning Vinca species, as well as based on our experience concerning plant extracts. When it comes to alkaloids, we have included all standards that were at our disposal besides vincamine and vinblastine, as one of the main purposes of this manuscript is to assess alkaloid content is four Vinca species.
Q7. Some lines must be added in the introduction section, with more information about the types of molecules identified to date in the vinca species.
A7. As specified above, the following modification were made at lines 60-72 and 77-83:
“The first isolation of a Vinca alkaloid was made by Lucas in 1859, and since then the research continued along the centuries [26]. In the early 1900’s, Vinca species were still insufficiently studied, and three new alkaloids were described for V. minor and three for V. herbacea [27-29]. Later, 43 alkaloids were isolated from V. major, 40 from V. minor and three from V. herbacea. After 2010 and up until now, new alkaloids are discovered and isolated almost every year only for V. minor, nine new alkaloids were described and isolated from V. major and two from V. herbacea [19,20,27,30,31].”
“Along alkaloids, other important natural compounds found in Vinca species are phenolic acids, carotenoids, caffeic acid, iridoids, flavonoids, amino acids, and other polyphenolic compounds [16,40]. One polyphenol that seems to stand out in high amounts throughout Vinca family is chlorogenic acid, considered a marker for leaf epidermis metabolites[41]. Other polyphenols have also been identified in various Vinca species, in far lesser concentrations compared to chlorogenic acid, such as p-coumaric acid, caffeic acid, ferulic acid, rutin and kaempferol [42,43].”
Q8. I miss in the M&M section the description of the rutin equivalent antioxidant capacity (REAC).
A8. Rutin Equivalent Antioxidant Capacity is in fact ABTS bleaching assay with rutin as standard. In order to avoid any vagueness, we have also included this mention in the text.
Q9. Lines 488-495: do this part describe the “Induced peroxidation of liposomes (LipPx)”? Please specify
A9. We have included the full name as well as the abbreviation in the M&M section to make it clear for future readers.
Q10. Figure 1, Figure caption: “the identified compounds are marked in green”. Do authors mean the compounds identified in the Vinca samples? Please, specify
A10. Yes, we mean the identified compounds in the Vinca extracts studied. We have modified the text accordingly to make it clearer.
Q11. Table 1: vinblastine was not detected in any of the vinca species. Why is it listed in the table? Since the other not detected compounds are not listed in the table, it should be removed.
A11.We included vinblastine as it is a molecule related to vincamine and expected in plants rich in indole alkaloids. We have removed it from the table, as suggested.
Q12. Tables and figures captions and footnotes. What type of statistical analysis the symbols “*”, “**”, “***” refer to? More in general, please add in the material and method section a paragraph with detailed information about the performed statistical analysis.
A12. The information was added at lines 576-582 and to each table and figure caption that had statistical analyses performed.
Q13. Lines 131ff: in the material and methods sections, authors state that, starting from 20 commercial standard, they built calibration lines. Consequently, I would have expected to find in the manuscript quantitative data about the identified molecules and not the sums of areas of peaks related to each category of compounds. Even for those peaks not attributed to a specific molecule, but included in one of the four categories (flavonoids, cinnamic acids, hydroxybenzoic acid, alkaloids), a quantitative datum can be easily calculated using the calibration curve of a standard belonging to that category. This part has to be definitely improved, and the other part of the manuscript, in which quantitative data from HPLC analysis have been used for example for identifying correlations, have to been changed accordingly.
A13. The reviewer is right. We changed accordingly in the manuscript (lines 151-152) and in the involved figures (Figure 2e and Figure 3).
Thank you for your support!
Reviewer 2 Report
The aim of the study was to compare the obtained extracts of Vinca L. speciest leaves in terms of their antioxidant, antibacterial properties and cytoctoxic effects. The topic of the work is interesting, the number of determinations made on the extracts is very large. The analytical methods are standard, but I suggest that in the future LC-MS / MS methods should be used, which would allow for the precise identification of compounds contained in the extract.
Detailed comments:
The methodological part of the manuscript lacks a description of the experimental design. How much plant material was extracted for 3 days? How many repetitions has the extraction been performed? How much extract was obtained? I recommend that this information be included in the chapter material and methods in subsection 4.1.
Was the method used to purify extracts from turbidity (filtration or centrifugation)?
The sentence ” The method was adapted after [72]” is incomprehensible (L528).
In the chapter describing the research methods, there is also no information on the methods of statistical processing of the results. The tables show the results of the statistical analysis, but it is not known which one. I recommend adding a chapter describing the statistical methods used, including the number of analytical repetitions.
The work also uses PCA analysis of obtained chromatographic peaks. What was the data processing program used? Has normalization been done, if so, by what method? I recommend that this information be included in the description of the statistical methods.
The tables use different terms for the extracts tested. In table 1 it is eg V. minor, in table 2 it is Vm. I propose to systematize the entries. Use the same abbreviations in all tables and figures.
How is it possible to explain the fact that for VMv and Vh extracts the total flavonoid content (TFC) was higher than the total phenolic content (TPC)? The variation in content is very high, so is there any other reason than the use of a different standard?
Author Response
Q1. The aim of the study was to compare the obtained extracts of Vinca L. species leaves in terms of their antioxidant, antibacterial properties and cytoctoxic effects. The topic of the work is interesting, the number of determinations made on the extracts is very large. The analytical methods are standard, but I suggest that in the future LC-MS / MS methods should be used, which would allow for the precise identification of compounds contained in the extract.
A1. We would like to thank the reviewer for their time given to read our paper and also, for the suggestion to improve our future work. Regarding the point-by-point indications given, they are answered bellow and in the main text of the manuscript, are marked in yellow.
Q2. The methodological part of the manuscript lacks a description of the experimental design. How much plant material was extracted for 3 days? How many repetitions has the extraction been performed? How much extract was obtained? I recommend that this information be included in the chapter material and methods in subsection 4.1. Was the method used to purify extracts from turbidity (filtration or centrifugation)?
A2. Thank you for the observation. Somehow, we overlooked these details but are now included in the main text at lines 438-446:
“During the flowering season (April-May and September) the fully developed leaves of adult plants were collected and thoroughly washed with tap and distilled water, cut into small fragments (0.5-1 cm), weighted, and placed in the percolator. The extraction of phytoconstituents was conducted as previously described [9], through the cold repercolation method in a 1:2 (w:v) solvent to fresh herba ratio. For three days the fresh herba was extracted with 70% ethanol (Merk, Bucharest, Romania), at room temperature. The fluid extracts obtained by filtration were as follows (w:v/g:mL): 1:1.2 (V. minor), 1:1.4 (V. major), 1:1.5 (V. major var. variegata), and 1:2 (V. herbacea). The final ethanol concentration was 30% in all obtained extracts.”
Q3. The sentence ” The method was adapted after [72]” is incomprehensible (L528).
A3. The sentence was rephrased:
“The method was adapted after an existing protocol [83] and”
Q4. In the chapter describing the research methods, there is also no information on the methods of statistical processing of the results. The tables show the results of the statistical analysis, but it is not known which one. I recommend adding a chapter describing the statistical methods used, including the number of analytical repetitions
A4. Thank you for the observation. A chapter with all missing information was included at lines 597-605
Q5. The work also uses PCA analysis of obtained chromatographic peaks. What was the data processing program used?
A5. STATISTICA 12 has been used for PCA analysis. Information now included in the text (597-605).
Q6. Has normalization been done, if so, by what method? I recommend that this information be included in the description of the statistical methods.
A6. Yes, min-max normalization has been performed for chromatograms before PCA application. We included this information in the revised manuscript (478-479).
Q7. The tables use different terms for the extracts tested. In table 1 it is eg V. minor, in table 2 it is Vm. I propose to systematize the entries. Use the same abbreviations in all tables and figures.
A7. Corrected as suggested.
Q8. How is it possible to explain the fact that for VMv and Vh extracts the total flavonoid content (TFC) was higher than the total phenolic content (TPC)? The variation in content is very high, so is there any other reason than the use of a different standard?
A8. The observation that the reviewer is making is quite a fine one as readers would expect the total phenolic content (determined using Folin-Ciocâlteu assay) to yield higher values compared to total flavonoid content (determined via complexation of flavonoids by aluminum in acidic environment).
A possible explanation, besides the difference in standards, can be drawn when inspecting the underlying working principles of each assay: while Total Flavonoid Content is based on complexation of flavonoids by Al3+ in acidic medium (thus rendering OH groups as very important; still, this assay is described in the literature as specific to flavonoids, so other factors that provide specificity are also involved- for further details see Pękal, A. & Pyrzynska, K. "Evaluation of aluminium complexation reaction for flavonoid content assay." Food Analytical Methods 7.9 (2014): 1776-1782), Total Phenolic Content (Folin-Ciocâlteu method) has its core at a process of oxidation of so-mentioned phenolic compounds. In the literature, as well as in one paper of ours (Zagrean-Tuza, C., et al. "Sugar matters: sugar moieties as reactivity-tuning factors in quercetin O-glycosides." Food & Function 11.6 (2020): 5293-5307), the fact that glycosides of the phenolic compounds have a higher redox potential is underlined; as showed in Table 1 in the manuscript, various glycosides have been identified by HPLC in Vinca extracts in relatively high amounts (especially chlorogenic acid, but rutin and quercitrin as well). Considering this, the presence of glycosides in Vinca extracts may lessen the values measured for Folin-Ciocâlteu, as the extracts as a whole are rendered less redox reactive towards the reagent used. On the other hand, the presence of glycosides might actually help in flavonoid quantification, as the principle involved is not a redox one, but the complexation at a metal center.
Thank you for your support!
Round 2
Reviewer 1 Report
The authors have correctly addressed all my suggestions. The paper can be published.
The only change I still recommend at this level is the follow:
Line 78-79: please change polyphenolic to phenolic and polyphenol to phenol here and throughout the manuscript.